# System dynamics model for an agile pharmaceutical supply chain during COVID-19 pandemic in Iran

Mohammad Hamzehlou *

Department of Management, Tehran University, Tehran, Iran

* hamzehlou@yahoo.com

**Data Availability Statement:** All relevant data are within the paper and Supporting Information files.

**Competing interests:** The authors have declared that no competing interests exist.

## Abstract

Unpredictable changes in the current business environment have made organizations pay attention to the concept of agility. This concept is a key feature to survive and compete in turbulent markets while considering customers' fluctuating needs. An organization's agility is a function of its supply chain's agility. The present study offers a System Dynamics (SD) model for Iran's Pharmaceutical Supply Chain (PSC). The model is presented in three steps. First, the Supply Chain (SC) indicators were extracted based on theoretical foundations and literature review results. Second, an SD model of the PSC was extracted in the context of the COVID-19 pandemic with the necessary analyses. Finally, the desired outputs and strategies were obtained by conducting a case study. The results indicated that the PSC's highest agility could be guaranteed by the simultaneous implementation of three strategies: investment, Human Capital Development (HCD), and accelerated completion of ongoing projects on a priority basis. According to these results, the organization had better determine the amount of capital and workforce required for ongoing projects, then design funding solutions to implement these projects and implement them according to the projects' priority.

## 1. Introduction

Nowadays, competition is essential in developing industries [1, 2]. Due to the intense competitiveness in the pharmaceutical industries, governments try to balance competition and economic growth [3, 4]. The high and continuously increasing drug costs are global concerns that require a PSC management system to manage [5, 6]. The supply, storage, and distribution of drugs are critical issues as most healthcare service receivers are outlying rural areas deprived of medical services due to large distances and extra. Thus, many of the population no longer needs to unreasonably visit specialized centers [7–10].

Interruptions in supply chains can occur due to natural catastrophes or failures. In the same aspect, numerous outbreaks of very contagious diseases such as the COVID-19 have caused a global catastrophe for humans and economies [11, 12]. The pandemic has greatly affected activities such as SC, manufacturing operations, logistics, and several other sectors [13–15]. Consequently, the epidemic has nevertheless affected the global economy. Many

countries have decided to execute a complete lockdown to control the spreading of the infection. Moreover, various governments have invested many resources to stop and control the spread of the coronavirus. However, the transmission of the coronavirus has made the struggles of the governments almost ineffective. While the authorities have attempted to contain the coronavirus outbreak, various organizations have also cooperated in this matter. Some organizations have adopted actions towards protecting their employees and implementing efficient care mechanisms. Unfortunately, many countries' inability to respond to the COVID-19 outbreak is due to the SC- the transportation of goods [16, 17].

To overcome the risks involved, organizations should plan carefully against uncertainty. If failed to meet customer demand, an organization would face significant consequences [18, 19]. These include reduced customer satisfaction, lack of self-confidence, pessimism, high inflation, and long lead time [20]. Hence, a Supply Chain Network (SCN) must be designed and planned to maintain agility against any disruption [21–23]. Moreover, due to the pharmaceutical industry's macroeconomics role, variables such as employment, economic growth, and non-oil exports are considered crucial [24, 25]. The supply and delivery of drugs involve some risks. For example, the product may be damaged, destroyed, not delivered on time, delivered by mistake (in terms of its type), delivered to another location, or sales slip errors. The customer may also not pay the prices or many other mistakes that can have far-reaching effects on the entire SC [3]. Purchasing managers react to the possible shortage of raw materials by placing large orders with suppliers [26, 27]. Some government regulations may affect importing raw materials and, consequently, producing the required equipment [28, 29]. Therefore, the need for a reliable SC in the pharmaceutical industry has been raised as a strategic plan. SC must be agile to answer all these needs [20, 30]. Thus, this study's main purpose is to provide a model for agile System Dynamics (SD) in the PSC. Due to the COVID-19 outbreak, there will be an increase in demand regarding consumer preferences, mainly due to fear and, consequently, changes in consumption patterns [21].

One of the main differences of this study is considering the status of pharmaceutical products and money transfer and payment conditions in the PSC. For this issue, more beneficial policies than previous studies can be presented by considering the variables related to payment methods and providing resources to cover these issues. Also, since there is a lack of comprehensive research, the proposed research answers the following questions related to the PSC of Iran:

1. How the information transparency affects the PSC agility during COVID-19?

2. How resilience affects SC agility during COVID-19?

3. What is the rate of prediction effects of new payment barriers on PSC during COVID-19?

System dynamics is an approach to understanding the nonlinear behavior of complex systems over time using feedback loops [31, 32]. The basis of the idea is dynamic systems. Scientists in the field of system dynamics believe that the evolution of any system is lawful and can be recognized and thus steer the path of evolution in the desired direction. The system dynamics approach aims to provide the analyst with the tools needed to discover these underlying rules [33]. The system's dynamics methodology not only claims to know the laws governing the evolution of the universe but also makes it possible to construct a model of real phenomena using a simulation tool. To a large extent, the simulated system has the characteristics of a phenomenon in the real world. System dynamics is a coherent approach to modeling that combines quantitative or qualitative aspects to simulate a phenomenon over time. What makes systems dynamic more appropriate than other methods for studying complex systems is the use of causal and flow diagrams [34–36]. Using an SD approach, this study examines the drug

distribution industry in Iran from an economic perspective and related dilemmas such as payment problems and the simultaneous occurrence of critical situations such as the sudden outbreak of COVID-19. This study comprehensively examines and demonstrates how to maintain SC agility in such conditions. To do this, we seek to maintain SC agility by implementing policies that affect agility variables, innovative payment methods, diversified/alternative supply and export sources, and cash reserves. Investing in companies supplying international raw materials that influence these companies' decision-making process in the occurrence of logistical risks arising from COVID-19 and political risks significantly increases supply chain resilience in decisions and previous commitments of suppliers. In other words, by investing in companies supplying raw materials, we can influence the decision-making process of these companies in special circumstances such as COVID-19. Some logistics risks from COVID-19 and even political risks can cause severe problems in the drug supply chains. It may be designed by competitors of the company or competitors of the host country. Such intelligent disturbances, which cause severe damage to the chain, can be remedied by trained and trained Human Resources (HR). In other words, intelligent and analytical HR can deal with specific political and logistical risks created by competitors or the host country by inventing new payment methods to suppliers. Thus, the development of human capital can increase the resilience of the chain.

A major part of supply chain disorders is in the supply of raw materials as well as product supply networks in international markets. Therefore, benefiting from diverse and alternative supply and export networks can increase chain resilience. Therefore, taking advantage of diverse and alternative supply and export networks can increase chain resistance. In other words, in special situations of logistical risks, such as COVID 19, having diverse supply networks in raw material-producing countries can increase chain agility. Also, the alternative export network can remove logistical and political constraints in such circumstances and increase the chain's resilience in delivering the product to buyers.

In light of the above, this study intends to guide SC members in identifying potential sources of risk by identifying indicators that affect the PSC, providing a model in this respect, and implementing appropriate strategies through a coordinated approach to reduce SC vulnerability. Hence, it helps them achieve the upper hand in the competition and satisfy customer needs. In this research considering the economic, political, and logistics risks, the process of presenting an SD model in the PSC was done in three stages. In the first step, the SC's indicators were derived based on desk research results on the theoretical foundations and research background. Then, we derived the conceptual model of pharmaceutical industry specialists from the PSC and analyzed the results. The third step eventually yields an SD model for the PSC system. The suggested SD method was selected to study changes over time and observe variables' effect on target variables and other variables. The SD included causal mapping and the development of computer simulations to understand system behavior. Finally, scenario options were tested systematically to answer a set of 'what-if' questions.

The rest of the paper is structured as follows. In Section 2, previous studies are reviewed, and variables affecting the model are extracted. Section 3 addresses the research method and the SD model. Section 4 proposed the scenarios and policies. Section 5 presents the managerial implementation. Finally, Section 6 concludes the study.

## 2. Literature review

This section will review previous studies in the Healthcare Supply Chain (HSC), COVID-19, and Pharmaceutical Supply Chain (PSC). The PSC plays a vital role in patient health due to its prominent role in this system. Hence, neglecting the role of PSC may affect patient health.

## 2.1. Pharmaceutical Supply Chain (PSC)

Consequently, the healthcare SC is an ecosystem consisting of organizations, individuals, technologies, activities, information, and resources that facilitate the manufacturer's cost-effective delivery of health products, vaccines, and other drugs [37, 38]. The primary purpose of healthcare SCs is to ensure rapid healthcare delivery systems to the citizens of a country, regardless of their geographical location, and respond to the patients' diverse needs [39]. A robust transportation and healthcare system cannot function well without a well-designed Supply Chain Management (SCM) system. It cannot ensure proper maintenance of essential health products to patients in need [40]. Healthcare SCs are growing and becoming more complicated due to increased corresponding sections and feedback loops in the system. Meanwhile, public health SCs face significant challenges due to rising patient expectations and the inefficiency of SC operations. This issue has increased interest in improving healthcare SC and patient services [20].

A better understanding of SC resilience is created by examining network characteristics [41, 42]. This approach allows developing new SC design models later and addressing disruptions and recovery over time [43, 44]. Additionally, by reviewing the SC resilience literature, the SC can be connected to other networks, including transport, command, and control, by modeling and focusing on resilience quantifications [45]. For this purpose, the history of using optimization simulation methods to design resilient SCNs is studied [46]. Results show the dynamics of resilient strategies and their different applications in SCs [47].

However, these strategies are not always fully used, as they are constantly criticized and should be examined [48]. Moreover, all of these strategies are associated with some risks, which must be considered to prevent SC disruption [19, 49]. One of the most significant risks is the lack of proper cooperation of suppliers. Such risks can be reduced through appropriate evaluation and selection of suppliers as one of the most critical factors [50]. It is also critical to identify different related risks, which can be dealt with immediately through this knowledge and awareness and the design of appropriate functional systems (e.g., decision support systems) [51, 52].

Paul et al., [53] develop a mixed-method approach consisting of both qualitative and quantitative techniques, namely online survey and the Best-Worst method. The empirical findings of this study show that increased food processing costs, lack of transparency and traceability, increase in the price of raw materials, lack of capital and physical resources, and spread of fake information are the top five sustainability challenges to the Australian food processing sector due to the impacts of the COVID-19 outbreak. Also, Eirill Bø et al., [54] investigate how the COVID-19 crisis affected delivery security and firms' preparedness and responses in Norway. Recognizing links, overlaps, and complementarity between the models, and using them step-by-step, we exploit synergies that enable more comprehensive assessments of strengths and weaknesses in firms' supply chains, covering gaps, prioritizing between improvement areas, and collecting input toward detailed, actionable risk mitigation actions. The finding of this study suggests that ongoing societal trends of facility centralization may add an element of vulnerability for firms while spreading important functions over multiple locations can ensure more operational flexibility.

Goodarzian et al., [55] displayed a new multi-objective multi-echelon multi-product multi-period PSC network accompanying the production–delivery–procuring–ordering–inventory holding-allocation-routing predicament under uncertain conditions. Moreover, Tat et al,. [56] analyzed a two-echelon PSC while considering a single supplier (pharma-supplier) and a single retailer (pharmacy) with one kind of stable shelf-life medicine. Consequently, Roshan et al., [57] discussed crisis management in PSCs, where three main objective functions were

considered to minimize the total cost of the network, minimize the unachieved demand, and maximize the fulfillment of social accountability.

Viegas et al., [58] reviewed three inquiries in each category: (i) How did the PSC reverse flows impact forward PSC processes? (ii) How were the reverse flows identified at the PSC stages? (iii) How can the reverse flows improve with the help of academic literature? Franco and Alfonso-Lizarazo [59] utilized a simulation-optimization approach based on the stochastic counterpart to optimize planned and effective PSC decisions. Zandkarimkhani et al., [60] presented a bi-objective mixed-integer linear programming model for creating a perishable PSC network with uncertain demands. The suggested model's goals were to concurrently minimize the network's total cost and the amount of lost demand. Jambulingam & Kathuria [61] claimed that the three PSC-level levels of process coordination were examined for the first time.

Mahajan & Tomar [62] studied the disruption of the food supply chains due to COVID-19. The results showed that the coronavirus had a considerable impact on the long-distance food supply chain, especially for urban consumers and farmers. Larrañeta et al., [63] found a high need for medical devices and personal protective equipment during the pandemic. Many people are still unprotected, as authorities were not ready for such a situation. Moreover, the study comprised three-dimensional printing and how it could tackle future pandemics and strengthen the SC. Varshney et al., [16] remarked on the impact of coronavirus on the Indian food markets and the shock it has had on the supply and demand of the SC. Han et al. [64] emphasized the crucial role of flexibility in SC disruption management in the current and future pandemics. Zhang et al,. [65] were the first to examine the impact of COVID-19 on the supply chain and the stock market.

## 2.2. System dynamics in supply chain disruptions

Olivares-Aguila & ElMaraghy [66] proposed a system dynamics framework to observe the SC behavior and assess disorders' impacts. In addition, Wang et al., [67] apprised the dynamic effect of the five scenarios about SC disruption. The authors figured out that the SC impacts disruption were commonly short-lived. Also, Sabahi & Parast [68] studied the firms' innovation and resilience related to SC. They found out that innovation helps firms reinforce capabilities that affect risk management capability. Song et al., [69] simulated the hypothetical pandemic effects on vegetable wastage and shortage. In another study, Ivanov [15] investigated the framework of a resilient asset for SCM for post- COVID-19.

Thilmany et al., [70] specified the factors that state why local responses to COVID-19 differ from the national dialogue on food SC disruptions. Also, El Baz & Ruel [71] studied SC risk management's role in mitigating the disruptions' effects on SC robustness and resilience in the COVID-19 outbreak. Also, Chowdhury et al., [72] provided a systematic review of the studies on the COVID-19 pandemic in SC. Nagurney [73] replied to the COVID-19 pandemic by creating SC network optimization models using significant variables, including labor and associated capacities. In addition, Kontogiannis [74] investigated the risk and its diffusion as viewed in an SD framework integrated by a resilience perspective.

## 2.3. Gap analysis

Today, the market is continuously influenced by environmental and external activities. An agile SC may not be the least expensive SC but can overcome uncertainties and unexpected events in the business environment. The concept of SC agility expresses a multidimensional phenomenon where most SCs are threatened by various risks that cause disruption. This issue is particularly sensitive in the supply of drugs and medical equipment. Additionally, the SC also can restore pre-disruption conditions or even create better conditions. This feature is the

reason for the need for agility in the SC. The process of providing a model of agility in the PSC is done by taking economic, political, and logistical risks into account.

In the HR sector, we have focused on the cooperation between the parties involved in the chain and the financial sector on payment and related innovations due to Iran's financial conditions. Considering these issues and their impacts on the drug supply chain agility, this paper considers different conditions over time in the form of three scenarios by designing different policies in these conditions. The reason for choosing the system dynamics method in this project is to examine changes over time and observe the effect of variables on each other and the target variables. System dynamics include causal mapping and the development of computer simulations to understand system behavior. According to the above, the contributions of this research are as follows:

- Planning a set of goals and establishing a link between strategies and evaluation criteria

- Determining important and influential indicators on agile PSC

- Investigating the effect of information transparency, PSC resilience in the COVID-19 pandemic, and new payment methods barriers.

- Define four policies based on financial and HR issues and related issues such as business partners' participation that improve access to these resources or implement different projects to achieve these two important issues.

- Investigating the effect of quadruple policies with three defined scenarios to obtain the PSC agility in the COVID-19 pandemic.

- Considering the sanctions conditions of PSC companies in Iran and its impact on agility based on the parameters of the level of cooperation among members of the SC, government regulations (due to sanctions and the diplomatic situation), and cash resources for the supply of goods.

## 3. Methodology

System dynamics is the method utilized in this study to analyze and manage complicated feedback systems, such as those seen in the world of business and other social systems. This technique may be used to many different types of input [75]. The dynamics of the system cannot be utilized to investigate any system since feedback is the key characteristic of a system that has been studied in this way.

Creating feedback loops provides information on a strategy's performance and potential consequences in the system dynamics method [76, 77]. The conclusion drawn from the causal loop diagram's positive and negative loops is that changes in one element in a cycle may be identified as having an impact on another factor, leading to a greater knowledge of the system's performance and feedback. The outcomes of policy implementation in the future on the indicators are examined, and changes are assessed, once the model is complete, the conditions of a policy are applied, and the policy is put into effect [78, 79].

System dynamics has a number of advantages, including the ability to create complex models with numerous equations, the ability to track how the system works, the ability to display feedback, a high level of understanding of causal loop relationships, and ultimately a reduction in managerial decision-making errors by adhering to causal loop relationships, simulating actions in the model, and seeing potential outcomes [80, 81]. It should be highlighted that system dynamics has its own applications and that perfect answers to issues should not be sought, since this might lead to misunderstandings and negative feedback. This strategy's primary goal

is to provide a virtual laboratory for the evaluation of various policies, followed by the deepening and expansion of managers' understanding of the cause-and-effect relationships in systems [82, 83]. The system dynamics approach was chosen to address the issue since the study topic is complicated, there are numerous links between the various components involved, and some of the relationships between the factors are ambiguous [84–86].

Today, there is no doubt about the importance of medicine in the public health system. Purchasing managers react to the rumors of a possible shortage of some raw materials by placing large orders. This issue leads to a false increase in demand and, consequently, a real shortage. Some government regulations may affect the import of raw materials and, therefore, the production of equipment. Production may also be disrupted by repairing equipment, obtaining licenses or standards, as well as disruptions caused by natural disasters. Meanwhile, the mentioned risks are intensified by Iran's political and economic sanctions and the resulting instability [75, 83, 87]. To reduce these risks, the SC should be designed to be ready to deal with events, provide an efficient and effective response to them, and recover from pre-disruption conditions or even establish better conditions. Accordingly, the SC must be agile [7, 88, 89]. Hence, this study's primary purpose is to provide a model for an agile SD in the PSC.

Iran Pharmaceutical Group Company (Public Joint Stock Company) with more than 20 subsidiaries was established in 2010 based on Iran's pharmaceutical industry's analysis and foresight under the name of "Iran New Pharmaceutical Technologies Company" to meet the society's needs. The company, as a knowledge-based economic complex, has chosen its approach to focus on health products subsidiaries of Iran pharmaceutical holding operate in the fields of development and transfer of technical knowledge, commercialization of technologies, and drug production. This collection provides 14% of the country's total essential medicines in the COVID-19 pandemic conditions and produces more than 420 items through its subsidiaries. According to its macro-strategic policies and development plans, Iran Pharmaceutical Group Company concerning the analysis of the pharmaceutical market of Iran, the region, and the world to achieve its goals and respond to the country's needs and strategic analysis of the pharmaceutical industry. Therefore, in this regard and to help these strategies, we have presented a dynamic model of its supply chain system, using the last ten years of this company's modeling data. The results of these studies are expected to strengthen the indicators affecting agility for use in SC units of this industry and using these results. The company can improve its SC agility, the risk of SC failure in conditions such as reducing the massive volume of COVID-19 during rapid distribution. Therefore, the company's experts should know the indicators affecting SC's agility and implement essential strategies. This research uses a causal model to identify these indicators and the relationships between them, design a stock-flow model, and validate and implement different scenarios, smoothing the way to implement an appropriate scenario. The following steps were gone through in this study for a good SD.

### 3.1 Ethics statement

This study after reviewing all results was approved by the editorial board member of the management department of Tehran university.

### 3.2. Study flowchart

This study investigates the application of the SD model in the Iranian Pharmaceutical Group through the following steps. In the first step, the factors affecting the agile SC in the pharmaceutical industry were identified using the library data and information of experts in the industry and the Iranian Pharmaceutical Company. The second step examined the relationship between these factors, their interaction, and their type, considering whether their direction

was positive or negative. Once the experts finally confirmed the identified effects, the resulting information would be used for the sake of review and analysis. The analysis results might be applied for all SCs to identify the factors affecting these systems and optimize them. The system and its resources could be optimized, and their efficiency could be increased while considering these factors.

## 3.3. Developing a dynamic hypothesis

A dynamic hypothesis is an explanation of the conditional reference behavior that must be consistent in the model. A designer uses a dynamic hypothesis to extract and test the consequences of feedback loops. A series of diagrams will then be drawn to show the main mechanisms that stimulate dynamic system behavior. A model cannot be developed without understanding feedback loops. An excellent dynamic hypothesis and a well-known primary mechanism mean enough information for the system to start presenting level and rate equations. So, it is time to perform the next step of the modeling process, namely, formulation [90, 91]. Talking to the experts and reviewing previous research revealed that some criteria are more effective than others. Thus, they were omitted to simplify the model through similar effects. In this study, three primary hypotheses were defined:

1. **Hypothesis 1:** By designing different policies and observing the system's behavior, the optimal value for the model state variables can be obtained when different scenarios occur and by implementing appropriate policies.

2. **Hypothesis 2:** Increasing cooperation, group partnerships, and developing human capital system agility.

3. **Hypothesis 3:** The appropriate path in an agile PSC can be identified by designing an SD model.

## 3.4. Model structure

**3.4.1. Defining the model boundaries.** One of the most basic modeling concepts is defining the model boundaries and restricting them to a specific area for analysis and planning. Influential and impressive factors are located in this zone. Significant but less impressive factors are located outside the system boundaries. Finally, less significant and less impressive factors are removed from the model. Since such delineation severely limits the system, experts believe that developing a systematic perspective is necessary to properly observe and determine the influential factors and their relationship to the environment [92]. Table 1 demonstrates the key variables in modeling the research problem.

Previous research and expert views were used to compile these variables. The most significant difference between them and other studies is that one or more variables have been employed in earlier studies. In no research has ever employed all these variables at the same time to investigate their impact. We did not find any research on these variables' impact on each other as a whole. This issue has been investigated in this study utilizing SD.

**3.4.2. Model loops.** By incorporating all significant items in the research and eliminating similar items by experts, the final items were listed in Table 1. The relationships between these variables were determined, whether inputs or outputs and represented using a stock-flow diagram. Finally, the resulting model was validated through the tools that would be mentioned in the next step. Feedback loop thinking focused on identifying booster loops in the system and balancing loops that limit the growth. This study used the cause-and-effect method and, finally, its conversion into a stock-flow diagram to model and display the relationships

**Table 1. Key variables affecting an agile PSC.**

| No. | Indicator | Description | Type |
|---|---|---|---|
| 1 | Reaction rate | Rate of change of status according to the circumstances | Endogenous |
| 2 | Response time | Response time from request to delivery | Endogenous |
| 3 | SC agility level | Rate of reaction to changes | Endogenous |
| 4 | Common planning | Joint planning and coordinated decision making | Exogenous |
| 5 | Cooperation level between members | The degree of participation of the activities of the departments and individuals of SC in one context | Endogenous |
| 6 | Existing IT between members | Using information technology to circulate information | Endogenous |
| 7 | Sharing information | Information and its circulation along with SC | Endogenous |
| 8 | Creating a culture of honesty and friendship | Managing member relationships and the culture of honesty and truthfulness among them | Endogenous |
| 9 | Manage communication between members | Necessary integration in behavior and decisions | Endogenous |
| 10 | Trust between members | Behavioral interaction between members | Endogenous |
| 11 | Create a risk management team | Existence of risk management team to review risks | Endogenous |
| 12 | Compatibility | The ability of SC to adapt to environmental conditions | Endogenous |
| 13 | Ability to respond to disturbances | The power to deal with disruptions and unforeseen events such as the COVID-19 pandemic | Endogenous |
| 14 | Identifying and understanding the SC | Understand all SC processes | Endogenous |
| 15 | The technical know-how of the SC | Having technical knowledge in the field of SC implementation and execution | Endogenous |
| 16 | Structure | Structure and context of SC implementation and execution | Endogenous |
| 17 | Integrated Financial Resource Management (IFRM) | Harmonized system of using financial resources | Endogenous |
| 18 | Transparency of demand information | Identify demand and understand its amount | Endogenous |
| 19 | Access to financial information (budgeting) | Availability of budget information for members | Endogenous |
| 20 | Transparency of SC scheduling | Precise SC planning for all members | Endogenous |
| 21 | Government regulations | Government laws governing the drug distribution industry | Exogenous |
| 22 | The technical know-how of human resources | Mastery of human resources in SC technical knowledge | Exogenous |
| 23 | Supply and distribution strategy | Necessary policies and strategies for PSC and distribution | Endogenous |
| 24 | The level of risk management culture | Culture of cooperation to eliminate the risk | Endogenous |
| 25 | Member skills | Level of skills of members to implement and execute the SC | Exogenous |
| 26 | Relationship between members of an SC | The extent of the relationship between members, in terms of information exchange and division of labor | Endogenous |
| 27 | Interest in SC membership | Members' desire to join SC | Endogenous |
| 28 | Visibility | Have a clear view of all events and activities | Exogenous |
| 29 | Insurance regulations | Insurance laws governing the drug distribution industry | Endogenous |
| 30 | Commitment to contract execution | The degree of adherence and implementation of the points governing the contract | Exogenous |
| 31 | Prioritizing the use of financial resources | Use of financial resources to implement the project in order of priority | Endogenous |
| 32 | Cash reserves | Cash savings | Endogenous |
| 33 | Financing | Financing and providing capital for project implementation | Endogenous |
| 34 | Doing tasks in harmony | Performing activities by all SC activities | Endogenous |
| 35 | Proper circulation and transfer of information | Determine how information rotates in all parts of SC | Endogenous |
| 36 | Decision coordinating | Make decisions within a defined and coordinated framework between members | Endogenous |
| 37 | Sharing skill and experience | Sharing experience and skills among members to enhance efficiency and productivity | Endogenous |
| 38 | Various supply export sources/alternatives | - | Endogenous |
| 39 | Innovative payment methods | Discover and implement different and creative payment methods | Endogenous |
| 40 | Level of access to specialized human resources | The extent of SC specialists and their availability | Exogenous |
| 41 | Diversity of supply and export network | Using different networks to meet the needs and export of products | Endogenous |
| 42 | Rate of mastery of difficult conditions of competition and hostility | - | Endogenous |
| 43 | Rate of prediction of new payment barriers | Anticipate new payment barriers such as interbank problems and money transfers during sanctions | Endogenous |

*(Continued)*

**Table 1.** (Continued)

| No. | Indicator | Description | Type |
|---|---|---|---|
| 44 | Power to return to before of disrupted state | Ability to return to normal after a crisis and tension | Endogenous |
| 45 | Flexibility in supply, production, and distribution | Experience of SC, products, and distribution network against | Endogenous |

between influential factors. As mentioned earlier, the factors presented in Table 1 were examined. Next, the cause-and-effect relationships and, finally, the stock-flow diagram of the conditions, auxiliary, and rate variables were determined. From all-important model loops, six important ones are selected and shown in Fig 1.

First, Loop 4, the complete one, will be described. This loop also covers Loops 1 and 3. The only problem that causes loop 3 to be seen separately in this model next to loop four and individually is the flexibility variable's direct effect in the supply of production and distribution on the power return variable before the disturbance. This issue is not shown directly in loop 4. These two loops have shown that these variables simultaneously bring the model closer to the real situation.

High information transparency is required to increase SC agility due to the need to share information; therefore, information transparency increases with increasing agility. Russell & Swanson [93] speculate that SC scheduling transparency increases due to increased information transparency. This issue happens because other SC members can update their plans more precisely by being aware of each SC member's operational status. This subject ultimately leads to an increase in scheduling transparency. The greater the transparency, the greater the adaptability of members. This issue makes SC members coordinate their tasks more effectively and adapt by knowing the activities' timing process. SC Resilience tries to absorb and prevent the supply chains' capability to prevent and absorb alters and retake primary performance after disturbance [94]. According to this definition, the higher the adaptability, the greater the resilience of supply, production, and distribution. Moreover, the higher the resilience, the greater the ability to respond to disruptions. If agile, an SC can quickly adapt and respond to the disruption by eliminating it. Under these circumstances, it is clear that the higher the ability to respond, the greater the power to return to the pre-disruption conditions by resolving these disruptions.

According to the results of the research conducted by Da Silva et al., [95], Loop 2 illustrates that the higher the level of agility of the system, the shorter time, consequently, the faster the response. According to Jia et al., [96], Loop 5 shows that integrated management's financial resources increase with increased cash resources. So, this can be attributed to the availability of the required financial resource. The stronger the integrated management (due to increased financial resources), the higher the access to financial information (due to an improvement in the system). Furthermore, the higher the level of access, the easier it will be to prioritize using financial resources, as the importance of using them in any subject will be more transparent. At the same time, the information of the system is increased.

**3.4.3. Modeling time horizon.** Since the right time horizon plays a vital role in the SD outputs, the researcher will achieve more realistic results while considering the right time. According to Gray et al., [97], the effects of learner and feedback loops are generally not short-lived, while the cause-and-effect loops are short-lived. Hence, it is vital to manage a balance between these two. Moreover, Bakhshianlamouki et al., [98] believes that a 3 to 5-year interval is usually reasonable to review the results. Therefore, experts have considered a time horizon of 60 months (5 years) for SC agility to provide sufficient feedback performances.

**3.4.4. Cause-effect and stock-flow diagram.** This section presents the flow pattern by aggregating the experts' opinions of defined groups and eliminating duplicate factors. Then

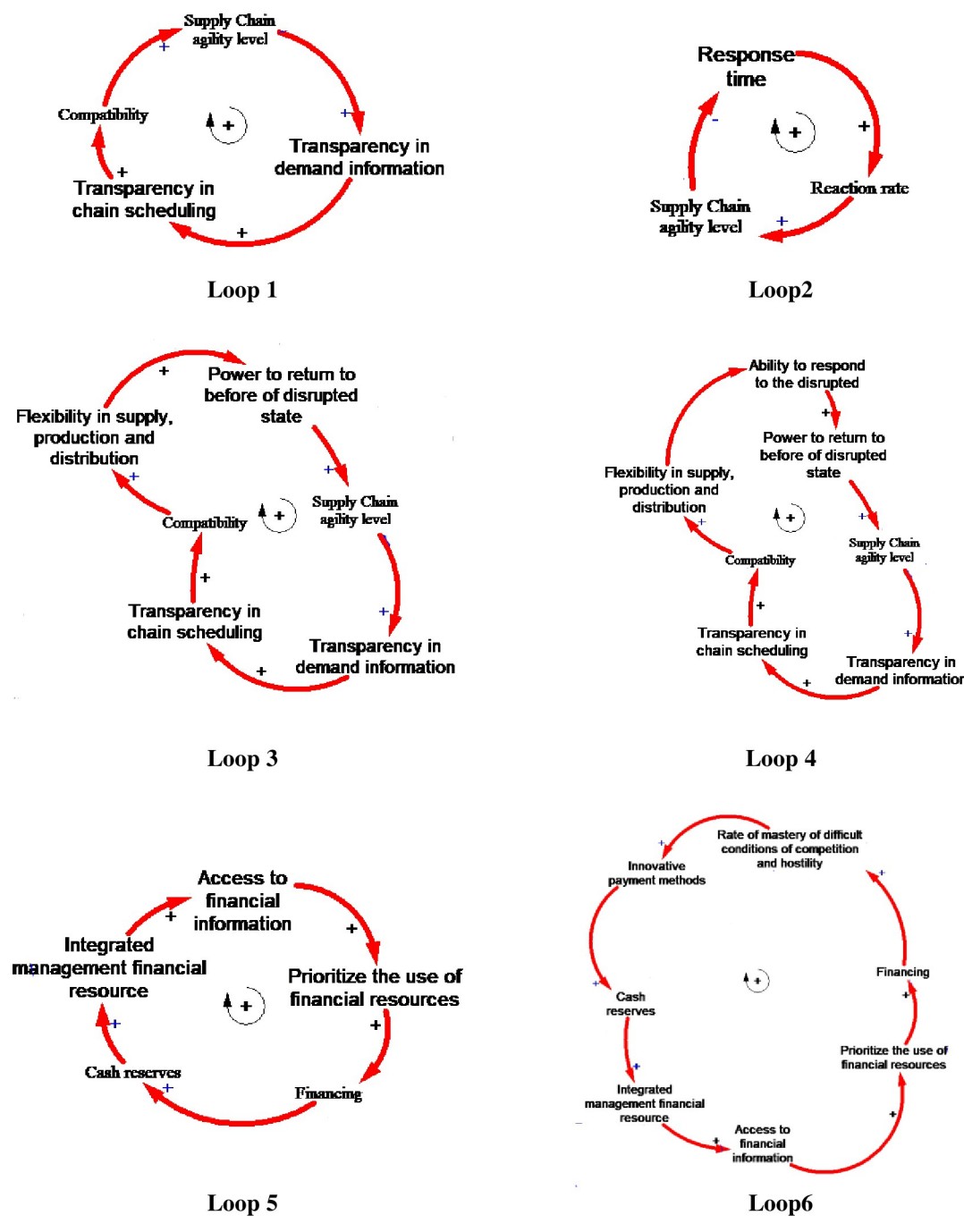

**Fig 1. Important model loops.**

some indicators were also defined as a subset of other groups, or an indicator was defined. Otherwise, it was described in another group. Thus, the SD model of this study (Fig 2) was developed after screening and reviewing all indicators.

In this study, the simulation period is estimated to be 60 months. According to the expert, the level of cooperation between SC members, risk management culture level, agility, cash resources, adaptability, and demand satisfaction time are among the most important factors

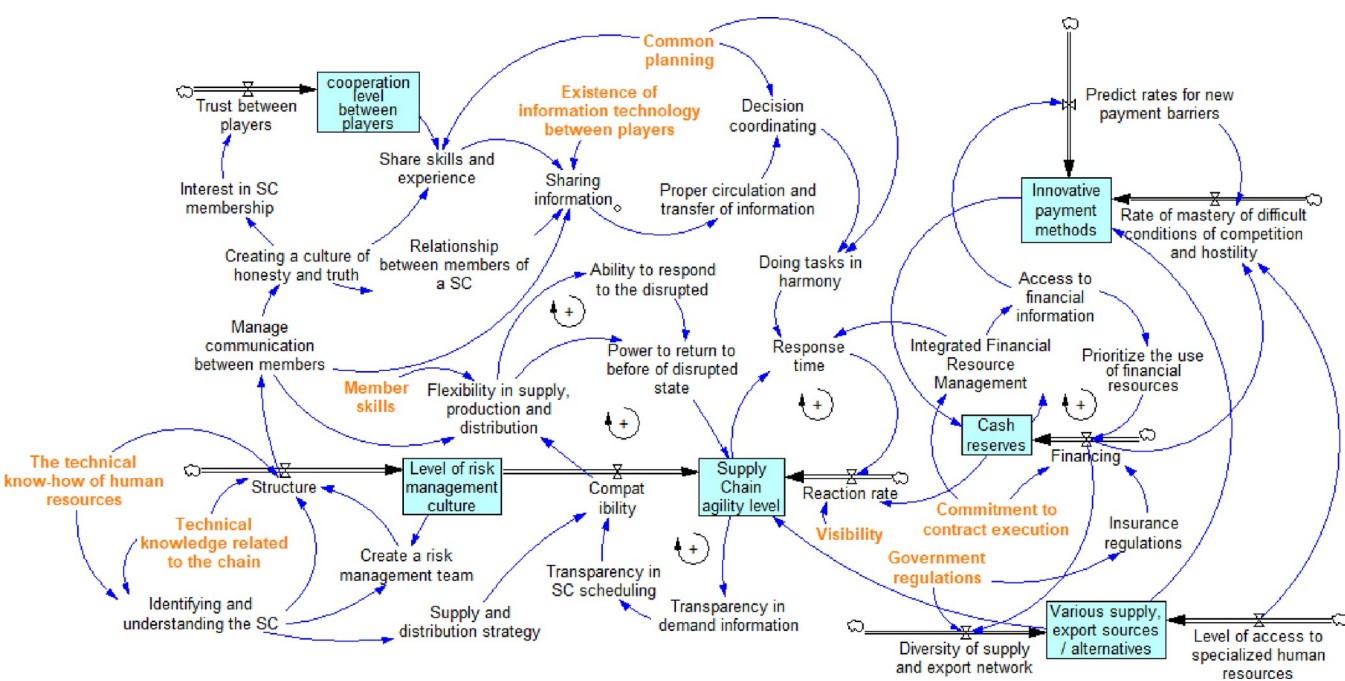

**Fig 2. Stock-flow model of the research problem.**

affecting the agility of the PSC. Transparency of SC information on demand leads to planning transparency. The more transparent the planning, the more prepared the system will be and, consequently, the more compatible. Increased adaptability allows the system to respond to the situations that arose under different conditions, prevents delays, and causes production and distribution resilience due to this readiness. This resilience will allow the system to return to its pre-disruption conditions, ultimately leading to SC agility. Moreover, HR technical know-how in the SC will increase SC's knowledge and help create an optimal structure. Properly structured, relationships between members are well managed, leading to resilience in SC, production, distribution, information sharing, and a culture of honesty and friendship. It should be noted that other factors, such as SC recognition and understanding, technical know-how of the SC, and the establishment of a risk management team, also affect the structure. Establishing a risk management team increases risk management culture, thereby affecting adaptability and leading to agility. On the other hand, recognizing and understanding the SC helps form an appropriate supply and distribution strategy to affect adaptability and agility.

The level of SC agility determines the level of demand information transparency. The more transparent the demand information, the more transparent the written schedule for the SC would be. This issue leads to greater adaptability and an increase in SC agility to create one of the system's feedback loops of the understudy. SC agility is enhanced by returning to the pre-disruption conditions, response speed, and SC reaction. The more transparent the SC issues, the faster the response to the consequences would be. The response speed depends on several other factors, such as available cash resources and response time. Response time is calculated based on SC agility, coordinated execution of tasks, and integrated financial management. Moreover, the availability of cash resources is in the feedback loop, which indicates the impact of cash resources on integrated financial management. This management allows access to financial information, prioritizes the proper use of financial resources, and enables capital and financial resources.

A higher SC agility level accompanies a more diverse/alternative supply source and different export sources due to greater accessibility and resilience. The supply and creation of diverse/alternative sources of supply and export are also affected by other variables. A higher rate of access to specialized HR leads to more diverse/alternative sources of supply. Also, it leads to an export increase due to an increase in the power to create and diversify resources using HR knowledge. This variable will sharply be increased by an increase in the diversification of the supply and export network, which can be attributed to wider access to various items. The rate of diversification of the supply and export network is affected by variables such as government regulations. For example, a ban on exports to a particular country under government regulations affects export diversity.

Innovative payment methods can increase cash resources. Cash resources will also change depending on the increase in the organization's dominance over tough competition and hostility. The more ways it provides paying through different payment methods (e.g., using third-party or other innovative payment methods). Furthermore, the development of innovative payment methods also depends on other factors such as the predictability of new payment barriers and the rate of access to specialized HR. As the predictability of new barriers increases, the need for innovative payment methods increases as well. Moreover, increased access to specialized HR is accompanied by growth in creating innovative payment methods.

Sharing information between SC members (i.e., sharing the technical know-how and information needed for various processes in the SC derived from information technology between actors), sharing techniques and experience, communication between members, and managing relationships between members help coordinate decisions. This issue leads to coordinated execution of tasks and also reduction of response time. Moreover, creating a culture of honesty and friendship helps increase SC membership and trust between them. This subject leads to increased cooperation between the members and, ultimately, the circulation and sharing of technique and experience. In general, it can be stated that:

- Agility is directly affected by the ability to return to the pre-disruption conditions, adaptability, and reaction rate.

- The resilience of drug supply, production, and distribution is affected by several factors such as adaptability, member relationship management, and member skills.

- The ability to respond to disruption and the resilience of supply, production, and distribution depends on returning to pre-disruption conditions.

- Information sharing stems from managing relationships between members, the relationship between SC members, the information technology available between members, and the sharing of knowledge and experience.

- Financing requires incorporating several factors, including insurance regulations, commitment to contract performance, and prioritizing using financial resources.

- Coordinated implementation of activities requires a joint program and coordinated decision-making.

- Response time depends on Integrated Financial Resource Management (IFRM), coordinated implementation of activities, and SC agility level.

This model considers some variables together: the variables "Level of Cooperation between members" with "Rate of Trust between members", the variables "Level of Risk Management Culture" with "Structural Rate", the variables "Level of SC agility" with "Adaptability Rate and Response Speed", the variables "Cash Resource Conditions", "Capital Rate", and "Diverse/

Alternative Supply and Export Mode" with "Supply and Export Network Diversity Rate" and "Rate of Access to the Specialized HR ", and the variable "Conditions of Innovative Payment Methods" with "Conditions of Dominance Rate in Difficult Conditions of Competition and Hostility" and "Rate of Predictability of New Barriers". In the Vensim software model, the relationships between the factors, the type of these relationships (decremental or incremental), and the feedback loops were specified.

**3.4.5. Model validation.** Validation of SD models is essentially a process to ensure the accuracy and usefulness of the model as a policy tool. The first step in determining the validity of a model is to judge its suitability for the intended purpose. This issue largely depends on the modeling's ability to intelligently perceive the signs of the problem and relate them to its causes. Judging how important a goal is essentially an art. The opinion of experts in the field under review can refine this judgment process. Accordingly, Sterman [99], Qudrat-Ullah & Seong [100], and Schwaninger & Groesser [101] proposed the tests such as Boundary Adequacy Boundary Conditions, Integrity Error, and Behavior Reproduction. These tests are performed based on the capabilities and structure of the model. Accordingly, in the implementation of these tests, there is no data as input. Therefore, it will be implemented only by changing the structure of the model based on the opinions of experts and will be compared and analyzed with the initial model.

## A) Boundary Adequacy Test

This test examines whether the selected factors influence the model or not. As previously mentioned, the factors studied in this research were determined by reviewing previous studies and experts' opinions [102, 103]. Therefore, the importance of these factors was confirmed by the two mentioned ways. The system behavior was evaluated in the next step by eliminating some of these essential factors to determine these influential parameters. After removing each of these factors, the model outputs could be shown as follows. Fig 3 presents the effects of removing the "Interest in SC Membership" factor on the variable "Level of Cooperation Between

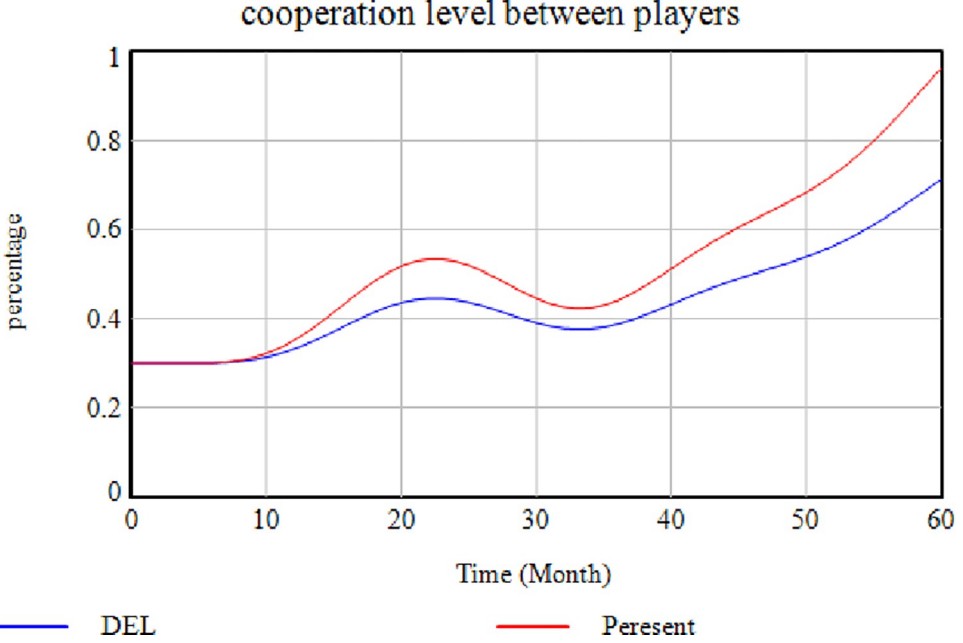

**Fig 3. Effects of removing the "Interest in SC Membership" factor.**

Members". In this case, the more the members are interested in membership, the more they will try to maintain their constructive cooperation in the SC, because they will not be removed from the SC. Hence, the direct impact of this interest on the level of cooperation is not deniable, as demonstrated.

According to this Fig 3, the cooperation level is higher where this variable exists (red diagram), while it decreases sharply when this factor is absent (blue diagram). This issue indicates a decrease in cooperation if the members are not interested in being in the SC. Fig 4 illustrates the effect of removing the "Establishing a Risk Management Team" factor. In contrast, the rest of the factors are assumed to be constant. In this case, the risk management culture level is reduced and maintained its distance from the "No Removal" mode in each case. The risk management culture can be enhanced by establishing a risk management team while developing risk science, promoting staff information, and providing methods for risk prevention (Fig 4). In Fig 4, the blue diagram shows how removing the "Establishing a Risk Management Team" variable affects the system.

Fig 5 depicts the effect of removing the "Power to Return to Pre-Disruption Conditions" factor. In this case, the level of agility of the SC decreases by removing this factor. According to the demonstration, the higher the SC's power to return to the pre-disruption conditions, i.e., the faster it responds to the disruption. The higher the level of SC agility would be. Agility means a quick response to environmental changes, while disruptions are the same as environmental changes. In Fig 5, the blue diagram shows how removing the "Power to Return to Pre-Disruption Conditions" variable affects the system. Fig 6 illustrates the effect of eliminating the "Insurance Regulations" factor on the "Cash Resource Level Conditions". In this case, the insurance companies impose high costs on the cycle and ask for a large part of the SC financial resources. According to Fig 6, the cash resource level is lower when this variable is removed (red diagram), while higher when this factor exists (blue diagram). This issue indicates an increased level of cash resources in the absence of insurance.

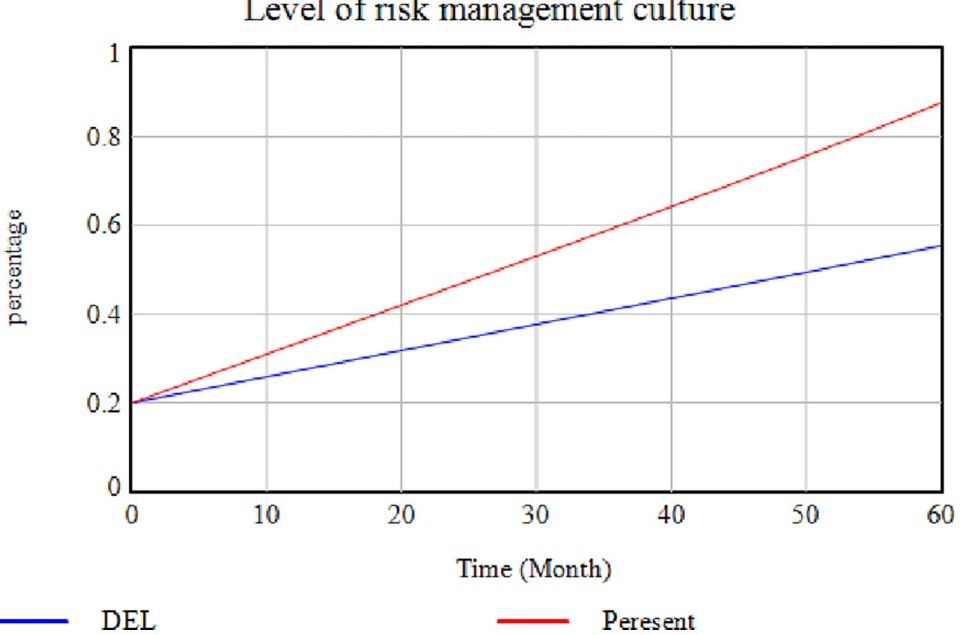

**Fig 4. Effects of removing the "Establishing a Risk Management Team" factor on the level of risk management culture.**

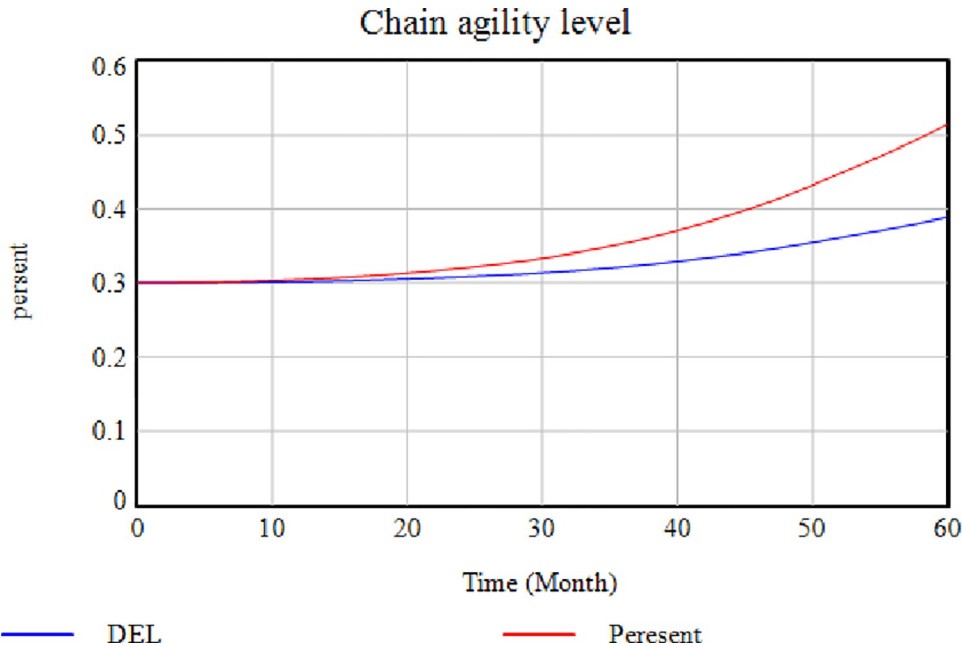

**Fig 5. Effects of removing the "Power to Return to Pre-Disruption Conditions" factor on the level of SC agility.**

## B) Boundary Conditions Test

This test examines the model's behavior when inputs are under boundary conditions, i.e., at their lowest or highest amount. It also checks whether the model is stable under such conditions. In the 'Boundary Adequacy Test' subsection, the status of the variables in the critical conditions (i.e., the maximum value) is examined in three conditions:

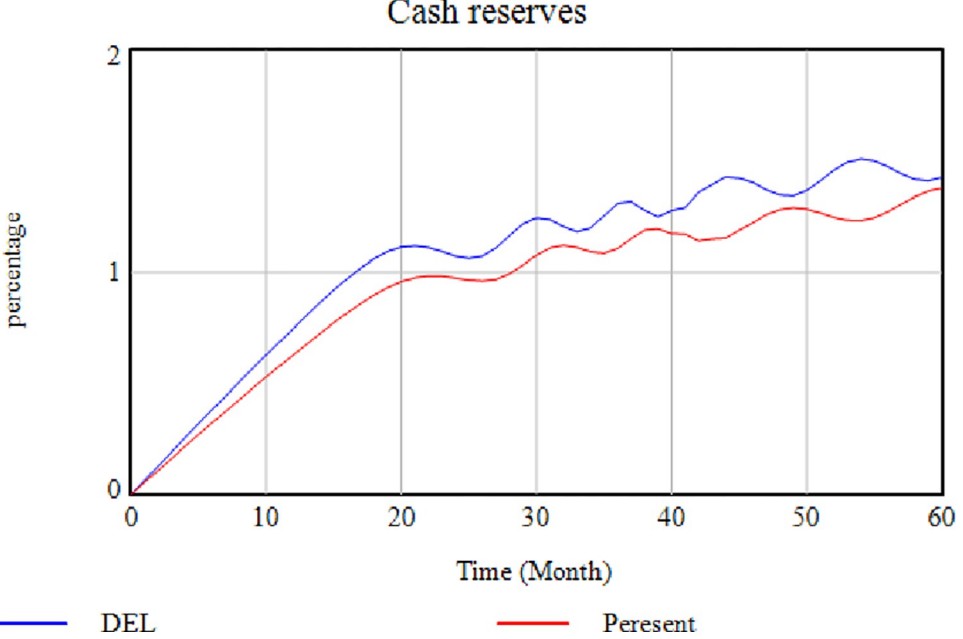

**Fig 6. Effects of removing the "Insurance Regulations" factor on cash resources.**

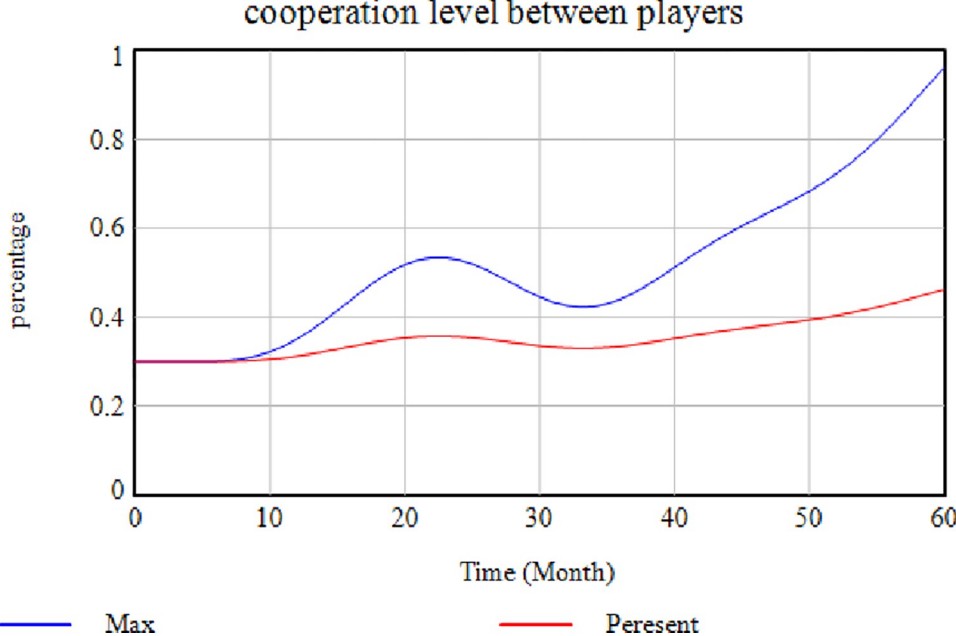

**Fig 7. Limit state of the degree of trust between members.**

- **Condition 1:** The trust rate between the members is at its highest amount (Fig 7).

- Complete trust between SC members allows them to perform their duties more transparently. This transparency in doing duties opens the way for cooperation instantly. The increased level of cooperation may initially make members tired after a short period due to the increased activity level, leading to a decrease in cooperation level. However, members will be more cooperative because they have realized the disadvantages of not having enough corporations.

- **Condition 2:** Knowledge and understanding of the SC are at their highest amount (Fig 8).

- High knowledge and understanding of the SC members help them recognize and understand the SC's risks more precisely. This knowledge allows the members to take a more specific path toward reducing risks and ways to manage them to reduce injuries in the event of an accident. Thus, as shown in Fig 8, the level of risk management culture (blue diagram) also increases when the SC recognition and understanding are at their highest amount.

- **Condition 3:** The reaction rate is at its highest amount (Fig 9).

   Agility is considered a desirable property under almost any condition, especially while talking about the SC. Agility can be defined as the ability to move quickly and easily and the ability to respond efficiently to opportunities and unavoidable threats. Therefore, the higher the reaction rate, the higher the agility, as shown in Fig 9. When the reaction rate is maximum (blue diagram), the SC agility level increases.

## C) Integrity Error Test

This test indicates the sensitivity of the model results due to time intervals. This test changed the 60-month interval of the model to 96-months. As shown in Fig 10, no changes were observed in the model behavior due to the model's time interval change. Factors affecting performance will continue to improve performance if they are controlled.

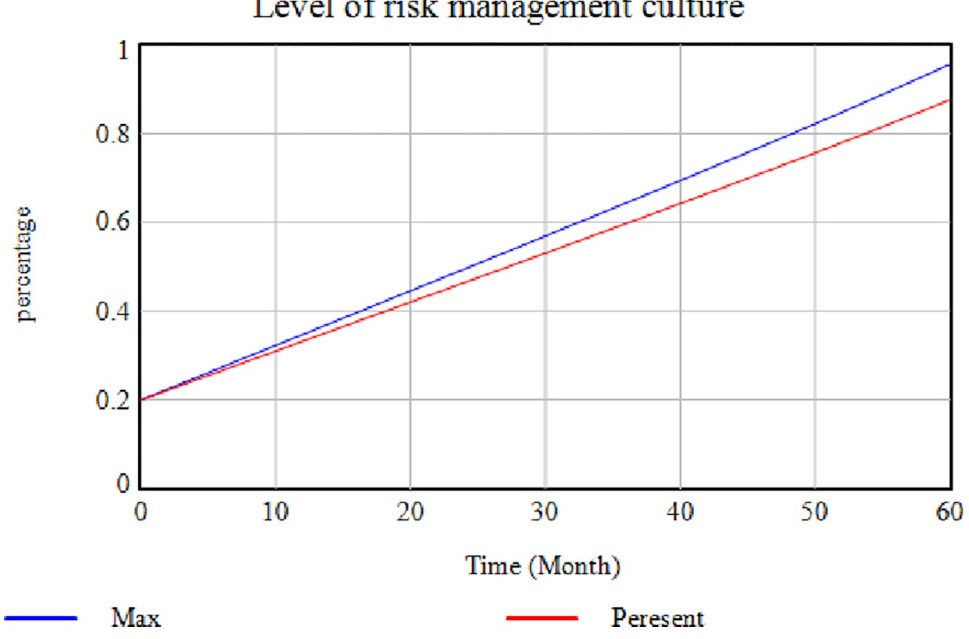

**Fig 8. Limit state of SC recognition and understanding.**

## D) Behavior Reproduction Test

Does this model reconstruct and display the system behavior under real-world conditions? The answer to this question is assessed using the Behavior Reproduction Test. According to the previous studies ' extensive investigations, the researchers believe this study includes variables that affect economic prosperity and Gross Domestic Product (GDP). Hence, it can

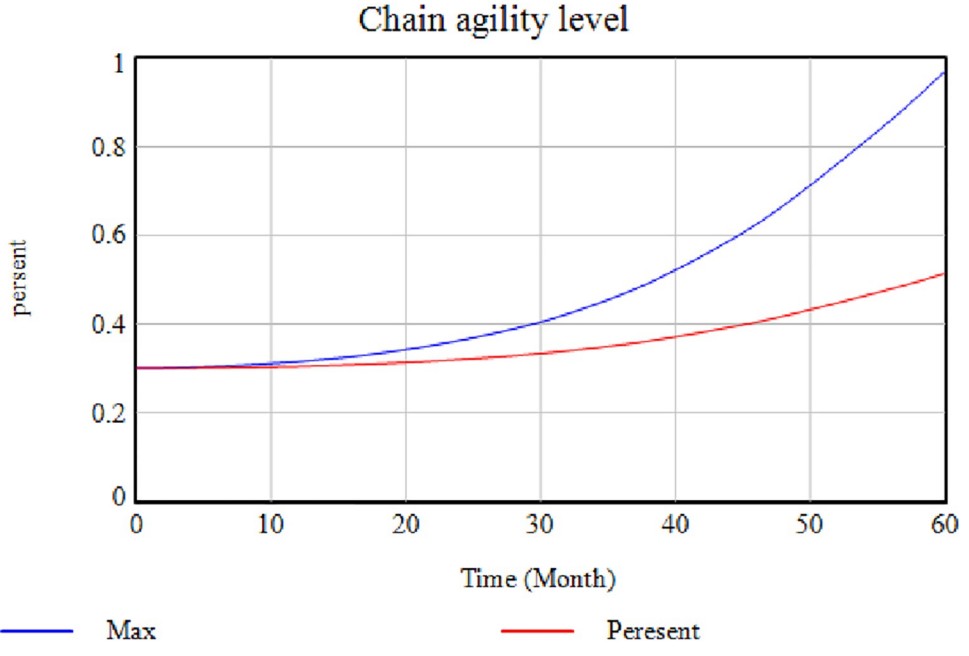

**Fig 9. Limit state of reaction rate.**

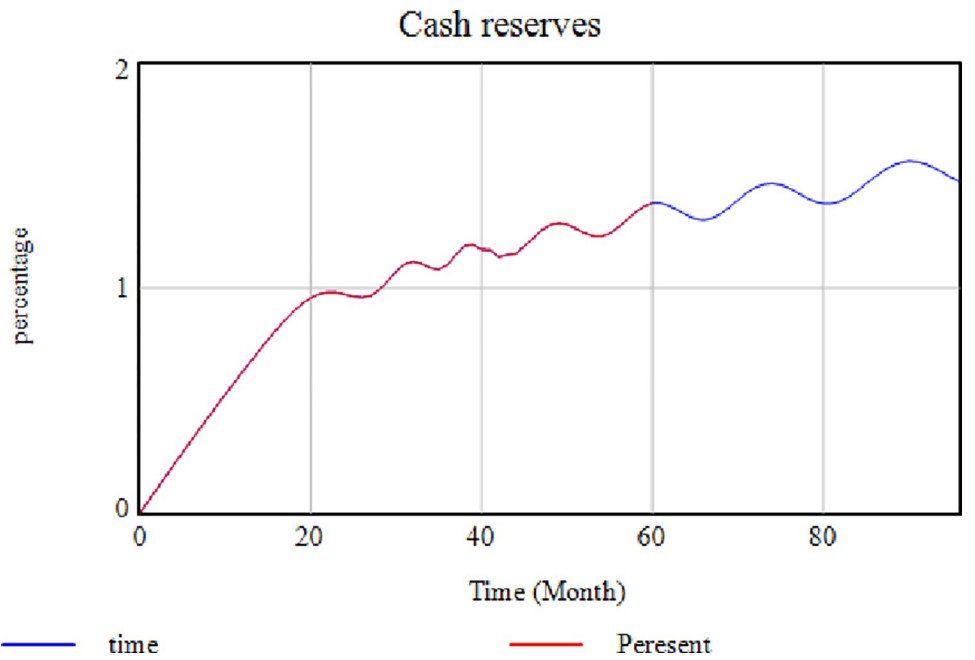

**Fig 10. Model outputs in the 60- and 96-month intervals.**

predict system behavior after identifying the corresponding criteria. Fig 11 shows that controlling factors affecting the creation of an SC can help increase SC agility. It should be mentioned that there are also other factors involved in achieving the desired level of agility in the SC of the Iranian Pharmaceutical Group, but the coordination of which requires more time.

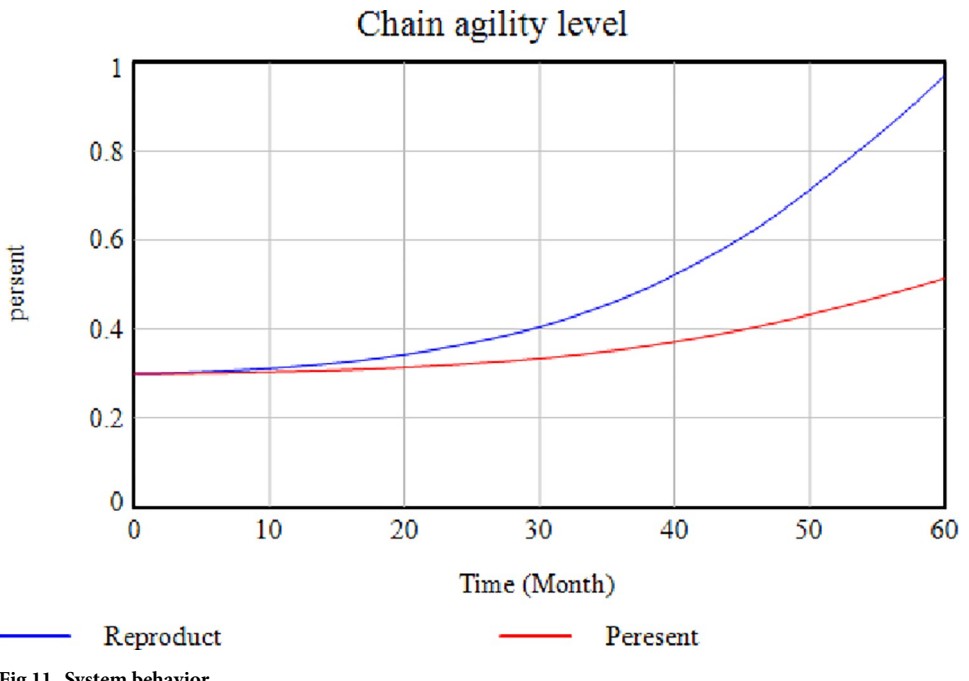

**Fig 11. System behavior.**

## 4. Simulation results

This section contains three subsections related to the simulation results. 1) scenario making, 2) policies, and 3) analyzing scenarios and policies are presented in this section:

### 4.1. Scenario making

After analyzing different variables and their impact on the desired periods' main variable, significant and decisive indicators on creating an agile SC in the Iran Pharmaceutical Group have been identified. The decision-maker enables us to choose different policies that include indicators to achieve effective strategies that affect the country's future (3 defined scenarios).

In this research, three scenarios have been considered and implemented for each stock variable. One of the possible conditions for significant indicators is considered using the opinion of the experts. Accordingly, Fig 12(A)–12(D) shows the model's status at the time of these scenarios. This study has drawn three scenarios for the PSC, considering its political situation and the COVID-19 pandemic. The study outlines possible future scenarios based on the embargo conditions and the conditions for discovering the COVID -19 drug.

In the first scenario, the sanctions against the pharmaceutical industry and the transportation and distribution industry increase daily. It is assumed that according to the current trend over five years, the sanctions will increase by 10%. With this happening, many variables in the model are affected. All the actors are not in the same country, then the cooperation between members will decrease. Government laws are increasing due to sanctions and the diplomatic situation. The rate of access to specialized HR, especially foreign specialists, is reduced. Innovative payment methods increase due to transfer money for medicine, labor, etc. Cash

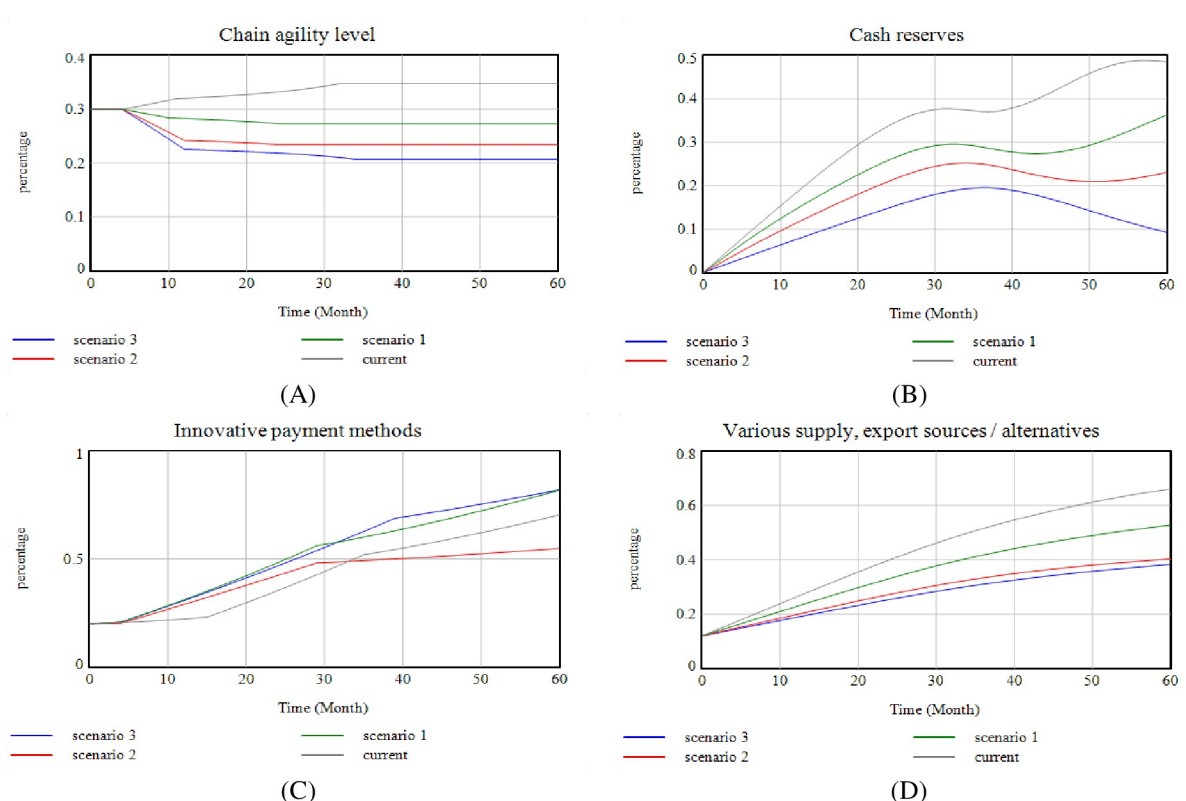

**Fig 12. Scenarios of the present study.**

resources are severely reduced due to the need to circumvent sanctions and increase costs and diversified/alternative sources of supply and exports are reduced due to sanctions and reduced transmission power. This scenario shows the worst-case scenario. In the second scenario, due to the global dependence on eliminating the pandemics, the sanctions will be stopped as they are, and the impact on the system will no longer be addressed. Instead, in this case, due to a sudden increase in demand, if not fully prepared, the reaction rate and the level of agility of the SC will be drastically reduced. Also, the response time will be increased, government laws are also reduced to respond quickly to society, and cash resources to supply goods are reduced. In the third scenario, where both the embargo and the discovery of COVID-19 occur, it is assumed that due to the involvement of the whole world with COVID-19, the rate of increase in sanctions against Iran will increase less rapidly. So in this scenario, sanctions increase by 5%. Due to these scenarios, it is assumed that access to medicine and equipment is difficult for patients with coronary heart disease due to political reasons and sanctions imposed in all sectors. But in scenario 3, it is assumed that due to the global epidemic of the spread of the disease is not limited to one part, if the disease is not controlled in all parts of the world, it will not be possible to stop the epidemic. Therefore, sanctions on drugs and medical equipment for Iran will be reduced to control the disease globally. With the reduction of sanctions on drugs and medical equipment, in this scenario, it is assumed that these changes have also affected the treatment requirements of coronary heart disease, and the process of sanctions is reduced. In these three scenarios, the model simulation for these scenarios compared to the current trend is shown in Fig 12.

## 4.2. Policies

Establishing the system dynamics model for policy simulation generally requires a series of steps. Two of the most important procedures are establishing causal loops to describe the logical structure of the system and building equations among the factors to generate quantitative relationships [104], which are illustrated below. Based on the existing studies about agricultural policy incentives and system dynamics, four main policies are proposed considering the specific circumstances of the Iran Pharmaceutical Group Company. Specific measures of these policies are shown in Table 2 and Fig 13. In addition, it should be pointed out that the 'current' curves in the following series of policy simulation diagrams indicate that none of the above policies has been implemented and that all parts of the model will develop as usual.

**Table 2. Scenario result (in percent).**

| Scenario | Policy | Agility | Cash reserves | Innovative payment methods | Various supply, export sources/alternatives |
|----------|--------|---------|---------------|----------------------------|---------------------------------------------|
| 1 | 1 | 29.5 | 23 | 85 | 58 |
|   | 2 | 27.5 | 35 | 97 | 60 |
|   | 3 | 29.8 | 30 | 99 | 65 |
|   | 4 | 31 | 42 | 97 | 67 |
| 2 | 1 | 22 | 12 | 62 | 45 |
|   | 2 | 23 | 16 | 66 | 50 |
|   | 3 | 26 | 14 | 70 | 55 |
|   | 4 | 31 | 9 | 72 | 57 |
| 3 | 1 | 23 | 8 | 76 | 40 |
|   | 2 | 24 | 15 | 81 | 47 |
|   | 3 | 31 | 14 | 93 | 50 |
|   | 4 | 33 | 18 | 99 | 52 |

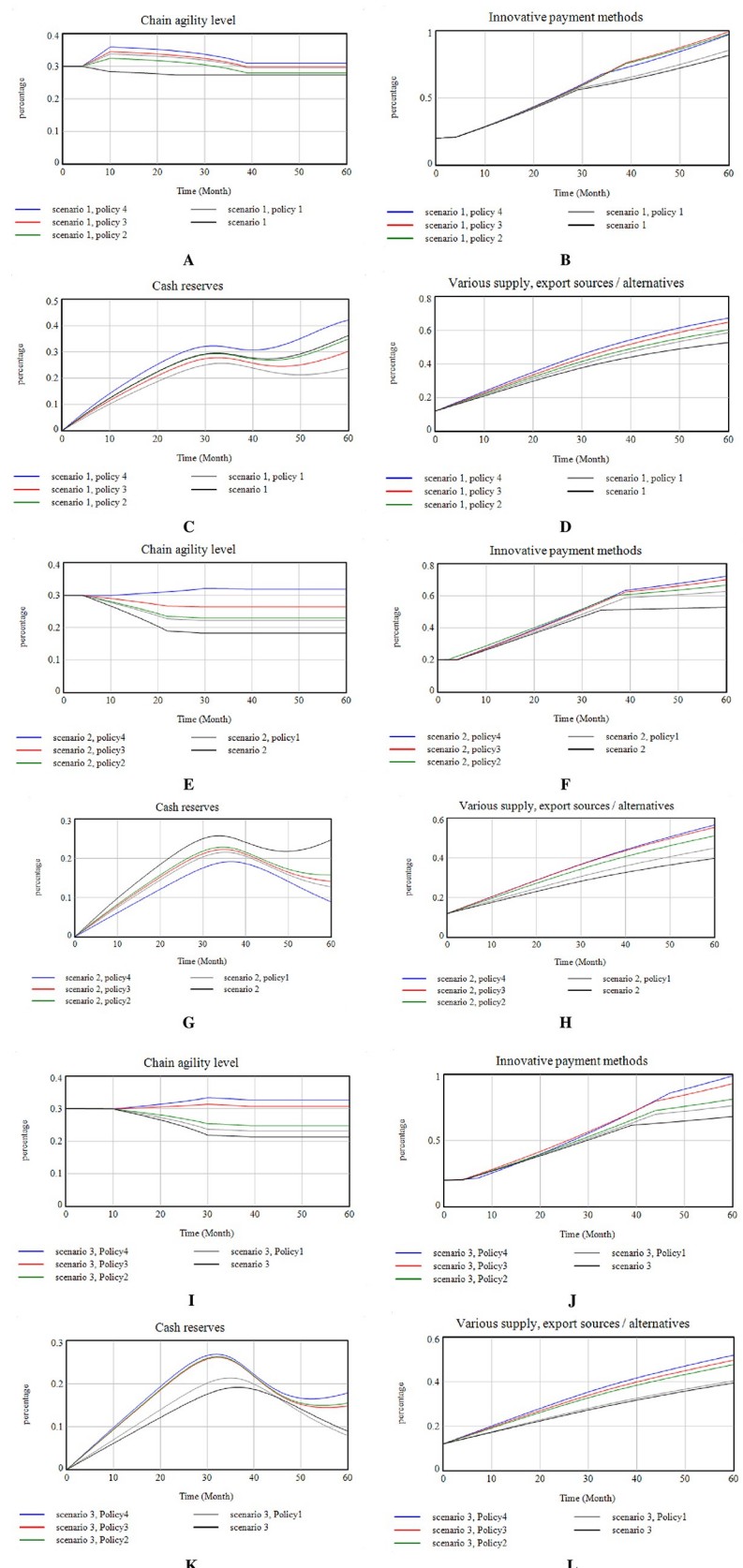

**Fig 13. Implementation of the policies of the study.**

Fig 13(A) shows that SC agility decreases steadily over five years as these scenarios occur. In this study, the following four policies are examined to show how these agility levels can be maintained and prevented if these scenarios occur.

- **Policy 1:** Increasing resilience in society requires large-scale investments, HR, and time: As mentioned earlier, apart from financial capital, the rate of access to specialized HR is also critical; nevertheless, it will not happen except through investments. Although there is a consensus on agility-enhancing goals, it is essential to invest in SC agility programs, individual initiatives, or projects, and strengthen weak infrastructures. This research sought to clearly show that investing in agility would bring significant and measurable profits to deal with this issue. Thus, this scenario is related to increasing investments. According to this company's plans, one of the plans is to upgrade the Good Manufacturing Practices (GMP) of the factories of Iran Pharmaceutical Group. The company is scheduled to allocate $100,000,000 for the investment budget in the next five-year plan.

- **Policy 2:** HCD: historically, SCs have emerged to meet diverse human needs, exploit natural resources, and empower people to engage in business. Studies show that three types of dynamic managerial capabilities are contributed to SC agility: 1) personal relationships with counterparts in supply companies (social capital), 2) public and private SCM skills (human capital), and 3) information processing style and the ability to understand SC disruptions (recognition). Blackhurst et al. [105] also noted that educated and well-trained employees are equipped with the skills needed to know when to take action. Human capital is a valuable resource needed to achieve organizational agility and create diverse/alternative supply or export sources. Also, under current circumstances and due to emerging problems, such as sanctions, innovative payment methods to deal with subjects like competition, hostility, and prediction of new barriers to payment are important and inevitable. This subject is just one example realized through specialized HR. This research model was based on variables, such as access rate to specialized HR, member skills, SC technical know-how, HR technical know-how, responding to disruptions, and recognizing and understanding the SC.

- **Policy 3:** This scenario simultaneously considers investing, developing human capital, and accelerating ongoing projects' completion on a priority basis. The previous two scenarios addressed the impact of investment and HCD. Optimizing and properly allocating financial resources based on investment priorities can significantly improve SC agility. Moreover, this prioritization of project completion and acceleration can help offer matched innovative payment methods.

- **Policy 4:** The last scenario considers increasing cooperation and in-group partnerships and the investment, HCD, and accelerating ongoing projects' completion on a priority basis.

## 4.3. Discussion on simulation results of all scenarios and policies

The implementation of these policies in the case of the three scenarios is shown in Fig 13.

As shown in Fig 12, if no new action is taken and the policy company continues its current policy, the following conditions will occur at the end of 5 years, depending on the occurrence of each of the three scenarios:

At scenario number one: Agility decreases from 30% to 27%; Cash reserves increase by 37%; Innovative payment methods increase by 55% and various sources, various export sources/options, increase by 40%. In scenario two: Agility decreases from 30% to 24%; Cash reserves increase by 23.5%; Innovative payment methods increase by 23%, and various sources, various export sources/options, increase by 40%. In scenario three: Agility decreases from 30%

to 21%; Cash reserves increase by 10%; Innovative payment methods increase by 55% and various sources, various export sources/options, increase by 38%. It is observed that if no new policy is created in the organization's program, agility will be reduced during five years, so to maintain SC agility, the mentioned policies have been implemented and simulated. The results in Fig 13 show what happens if any of these scenarios occur if these policies are implemented. Table 2 shows the results of the implementation of these policies. According to the results of the implementation of these policies at the time of occurrence of each scenario, it has been shown that implementing policies at the same time increase cooperation and intra-group partnerships, investment, HCD and accelerates the completion of ongoing projects based on organizational priorities. For purpose of the study, the agility of the SC will be maintained for 5 years, due to the stated benefits (increased inflation and the discovery of COVID-19), and even increased slightly.

Based on this, it has been shown that by designing different policies and observing the system behavior, the optimal value for the model state variables can be obtained when different scenarios occur and by implementing appropriate policies (Table 2). Therefore, Hypothesis 1 of this research has been fulfilled. On the other hand, it has been shown that increasing cooperation and group partnerships and HCD increases system agility (Hypothesis 2). In this way, the amount of information, knowledge, and experience shared between the parties involved increases with cooperation and group partnerships. With this increase and better decision-makers, affairs will be done in a more integrated and coherent way. This coherence increases system agility by preventing reworks. Finally, the results indicate that agility can be maintained in the long run by implementing the third policy in different conditions. By implementing the fourth policy, agility can be increased. As a result, it can be seen that by designing the model and examining the behavior of the variables in each of the possible conditions ahead, the appropriate path in the agile PSC can be identified and implemented (Hypothesis 3).

## 5. Managerial implementation

The shortage of drugs, supplies, and equipment needed to meet growing demand cannot be attributed to the COVID-19 pandemic. However, after drug discovery, pharmaceutical managers need to understand several weaknesses and SC dependencies beyond production. Pharmaceutical companies should also review their SCND, evaluate each step to identify potential failure zones, determine areas where more resilience should be provided, and provide the event agility. According to the present study results, the highest level of SC agility could be best achieved by simultaneously implementing three strategies: investment, HCD, and accelerating ongoing projects' completion on a priority basis. According to these results, to prevent SC failures when delivering COVID-19 drugs, organizations should start identifying, prioritizing, investing, and implementing their agility projects. Furthermore, according to this study's results, the required specialized workforce should have been provided to implement these projects and SCM in case of a possible shortage of COVID-19 drugs. In short, to handle the situation created in the interval mentioned above and to prevent the failure of the PSC, this research proposed the following recommendations:

- Determining and prioritizing programs required to achieve SC agility.

- Designing some methods for evaluating SC members. Evaluating these members' potential risks could be reduced at critical moments (e.g., COVID-19 drugs shortage).

- Investing in implement priorities

- Designing and implementing HR development programs and training courses

Finally, based on these results, pharmaceutical companies need to learn how to meet the ongoing challenges in their environment to ensure their survival and progress in the 21st-century. These results encourage pharmaceutical companies to adopt a new performance approach that enables them to be resilient and respond quickly to unpredictable changes. Thus, pharmaceutical companies must seriously apply SCM to resist unexpected disruptions in their SC to be successful. Finally, as corporate success is directly linked to answering customer needs, pharmaceutical companies need to pay close attention to their SC operations' agility.

## 6. Conclusions

The dynamic nature of a modern SCN has led to a great deal of uncertainty in many network parameters. Ignoring such uncertainties may pose several threats to the entire SCN. Organizations should carefully plan to control uncertainty to overcome these risks. An organization will face significant consequences if it fails to meet customer demand. These issues include declined customer satisfaction, pessimism, inflation, and higher lead time. Hence, an SCN must be designed and planned to maintain its agility against any disruption.

Moreover, the PSC is considered a major national category due to the pharmaceutical industry's role in macroeconomic variables (e.g., employment, economic growth, and non-oil exports). Most SCs are threatened by a variety of risks, leading to disruption. This issue is of specific sensitivity in supplying drugs. This study considered the risk of sanctions and an immediate increase in demand for COVID-19 as possible scenarios in the next five years. Section 4 shows that SC agility will decrease over time if these scenarios occur and there are no new policies, and the status quo is maintained. According to different policies, implementing policy 4, apart from maintaining agility, will ensure the highest SC agility level (33%) and lead to an agile system. These results show that to maintain and achieve the highest level of SC agility. The best decision is to simultaneously implement four policies: increasing cooperation and intra-group partnerships, investment, HCD, and accelerating ongoing projects based on organizational priorities. Therefore, this study shows that among the policies defined in the plans of Iran Pharmaceutical Group in the COVID-19 pandemic conditions, if the company's goal is to maintain SC agility when the stated scenarios occur, it should implement policy number 4.

The third policy was guaranteed the highest agility by comparing different suggested scenarios, financial resources, risk management, and cooperation between SC members. These results indicated that the highest degree of SC agility could be achieved by implementing three strategies: investment, HCD, and accelerating ongoing projects' completion on a priority basis. Given the current conditions of affairs in the world and potential conflicts at the time of the discovery of COVID-19 drug, and according to the results of this study, which focuses on determining and prioritizing programs required to achieve SC agility, future studies, may address the design of efficient, optimized models for SC agility in the pharmaceutical industry due to a possible shortage of COVID-19 drug. To this end, we first determine the factors affecting SC agility and, subsequently, the relationship between these factors and SC agility. Finally, we design the model by considering these factors and selecting the appropriate model.

## Supporting information

**S1 Data.**
(MDL)

## Acknowledgments

The author has done all the parts of this paper alone and has used the opinions of experts only in parts of the paper that are referred to in the text of the manuscript. Also, the author sincerely thanks the opinions of all the people who help in improving this scientific work.

## Author Contributions

**Writing – original draft:** Mohammad Hamzehlou.

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
