## [Decision Letter · Decision Letter 0]

14 Mar 2023

PONE-D-23-03622System Dynamics Model for an Agile Pharmaceutical Supply Chain During COVID‑19 Pandemic in IranPLOS ONE

Dear Dr. Hamzehlou,

Thank you for submitting your manuscript to PLOS ONE. After careful consideration, we feel that it has merit but does not fully meet PLOS ONE’s publication criteria as it currently stands. Therefore, we invite you to submit a revised version of the manuscript that addresses the points raised during the review process.

We look forward to receiving your revised manuscript.

Kind regards,

Saad Ahmed Javed, Ph.D

Academic Editor

PLOS ONE

Journal Requirements:

    "No"

5. Please ensure that you include a title page within your main document. You should list all authors and all affiliations as per our author instructions and clearly indicate the corresponding author.

Additional Editor Comments:

1. If possible, code should be provided to verify the results. One good way is to upload on GitHub or any other open access platform and then posting link in the paper for the readers.

2. A general graphical flowchart of the proposed approach should be presented and explained.

3. Comparative analysis is needed.

4. The complete specifications of the software and hardware systems used must be described.

5. A Table should be added where some studies should be reviewed related to your topic. Last row would be ‘The current study’ and show your significance against them.

6. The introduction must be improved. Experimental evaluation must be improved. Some improvements are needed in the description of the method.

7. It is important to clearly explain what is new and what is not in the proposed solution. If some parts are identical, they should be appropriately cited and differences should be highlighted.

Reviewers' comments:

Reviewer's Responses to Questions

**Comments to the Author**

1. Is the manuscript technically sound, and do the data support the conclusions?

Reviewer #1: Yes

Reviewer #2: Yes

2. Has the statistical analysis been performed appropriately and rigorously? 

Reviewer #1: Yes

Reviewer #2: Yes

3. Have the authors made all data underlying the findings in their manuscript fully available?

Reviewer #1: Yes

Reviewer #2: Yes

4. Is the manuscript presented in an intelligible fashion and written in standard English?

Reviewer #1: Yes

Reviewer #2: Yes

5. Review Comments to the Author

Reviewer #1: The study indicates a good research for the chosen topic and field. The article entitled System Dynamics Model for an Agile Pharmaceutical Supply Chain During COVID‑19 Pandemic in Iran (Manuscript Number PONE-D-23-03622) offers a broad theory and application according to academic requirements. The summary and introduction show a good theoretical approach. The research method and statistics are well studied. The results, discussions and conclusions show a good critical approach to the current research. The references indicate a good deepening of the studied field.

Reviewer #2: This paper is well organized, focusing on the construction of SD model for Pharmaceutical Supply Chain. Through the SD model, the authors present some management insights, such as the strategies for agile SC. My concerns are listed as follows.

The term should be consisted, such as Pharmaceutical Supply Chain (PSC) and Healthcare Supply Chain (HSC)

How to analyze the interaction effect of different policies on the system's behavior

How to better determine the amount of capital and workforce required for ongoing projects

6. PLOS authors have the option to publish the peer review history of their article (what does this mean?). If published, this will include your full peer review and any attached files.

Reviewer #1: **Yes: **Professor Cristian Delcea PhD

Reviewer #2: No

---

## [Author Response · Author response to Decision Letter 0]

25 Jul 2023

Dear Prof. Saad Ahmed Javed,

Please accept our sincere thanks for the valuable comments, which caused a proper revision in the paper, made it much more flowed, and brought a richer manuscript. We hope it is now up to your satisfaction and the high standards of PLOS ONE Journal.

The authors’ answers are colored in blue, while quoted texts from the manuscript are colored in purple. The reviewers’ questions and comments remained in black. The changes made to the manuscript are identifiable using the track changes mode in MS Word. The highlighted file just consists of major additions and omissions. For minor changes such as fixing punctuation, spelling, typing errors, using better words, suggesting better alternatives for meaningless sections, fixing grammar bugs, and so on, please refer to the track changed file, which contains almost all the changes made to the original submitted manuscript. In the highlighted file, the additions are marked in yellow. 

Best regards,

Mohammad Hamzehlou

Additional Editor Comments:

Thank you very much for the constructive comments, which undoubtedly resulted in an improved manuscript. Also, we appreciate your overall positive view of this research while considering a good potential for it. As a result of this revision, the original manuscript has undergone changes, the most important of which are detailed here.

1. If possible, code should be provided to verify the results. One good way is to upload on GitHub or any other open access platform and then posting link in the paper for the readers.

Response: We appreciate this valuable comment. We can't upload our equations on GitHub or any other open-access platform, but we can send the source codes of VENSIM software for more information and verification of results.

2. A general graphical flowchart of the proposed approach should be presented and explained.

Response: Thank you so much for this useful hint. We provided the graphical flowchart.

3. Comparative analysis is needed.

Response: We really appreciate this valuable and constructive comment. As you correctly mentioned, when it comes to expressing the results and discussing them, they should be compared with the previous ones mentioned in similar studies. In the original version of the manuscript, we tried to do so by comparing the results of this study with those of other studies conducted in this context, but because of being we should say in the system dynamics model is not applicable. Because when we create a model for a problem, we create a new and unique model. Also, we can compare the concept of each model only with the same model, not with different approaches such as DEMATEL or etc. Although methods such as DEMATEL are cause and effect based.

4. The complete specifications of the software and hardware systems used must be described.

Response: Thank you so much for your close attention to the text of the manuscript from beginning to end and for raising this useful comment. Also, we appreciate your fair judgment to mention that in the original version of the highlights, the objectives of the study were mentioned properly. To create the equations, researchers, based on previously existing data on the subject under study and using scientific methods, will extract the equations and use them in the relevant software (in this study, VENSIM). It should be noted that the use of these types of equations will allow researchers to use the required criteria in the model, if necessary. In other words, researchers are not limited just to predefined criteria. Accordingly, we added the VENSIM Software code for reader information and use.

5. The introduction must be improved. Experimental evaluation must be improved. Some improvements are needed in the description of the method.

Response: Thank you so much for this useful hint. We rewrote this section. Also, in some part of this section added more details. 

6. It is important to clearly explain what is new and what is not in the proposed solution. If some parts are identical, they should be appropriately cited, and differences should be highlighted.

Response: Thank you so much for your close attention to every single part of the manuscript and also for raising this useful comment. These added sentences and other minor changes made are traceable in the highlighted version of the manuscript too.

Investigating the effect of information transparency, PSC resilience in the COVID-19 pandemic

Designing and implement 4 policies affecting PSC agility based on financial capital, access rate. 

Investigating the effect of quadruple policies with three defined scenarios to obtain the PSC agility in the COVID-19 pandemic.

Considering the sanctions conditions of PSC companies in Iran and its impact on agility 

Review Comments to the Author

Reviewer #1: 

The study indicates good research for the chosen topic and field. The article entitled System Dynamics Model for an Agile Pharmaceutical Supply Chain During COVID 19 Pandemic in Iran (Manuscript Number PONE-D-23-03622) offers a broad theory and application according to academic requirements. The summary and introduction show a good theoretical approach. The research method and statistics are well studied. The results, discussions and conclusions show a good critical approach to the current research. The references indicate a good deepening of the studied field.

Response: Thank you very much for the constructive comments, which undoubtedly resulted in an improved manuscript. Also, we appreciate your overall positive view of this research while considering a good potential for it. As a result of this revision, the original manuscript has undergone changes, the most important of which are detailed here. We really appreciate the proposed hint. It is really nice of you to pay a close attention to every detail in the text. 

Reviewer #2: 

This paper is well organized, focusing on the construction of SD model for Pharmaceutical Supply Chain. Through the SD model, the authors present some management insights, such as the strategies for agile SC. My concerns are listed as follows.

• The term should be consisted, such as Pharmaceutical Supply Chain (PSC) and Healthcare Supply Chain (HSC)

Response: Thank you for proposing this comment. You are absolutely right. But in this research, we only examine the part of the pharmaceutical supply chain, that is, we examine how the members of the supply chain communicate for better efficiency and agility systems. Accordingly, the health supply chain is hidden in the above case.

• How to analyze the interaction effect of different policies on the system's behavior

Response: Thank you so much for raising this constructive comment, which has made the revised manuscript more readable. We added more details about it.

Action: At scenario number one: Agility decreases from 30% to 27%; Cash reserves increase by 37%; Innovative payment methods increase by 55% and various sources, various export sources/options, increase by 40%. In scenario two: Agility decreases from 30% to 24%; Cash reserves increase by 23.5%; Innovative payment methods increase by 23%, and various sources, various export sources/options, increase by 40%. In scenario three: Agility decreases from 30% to 21%; Cash reserves increase by 10%; Innovative payment methods increase by 55% and various sources, various export sources/options, increase by 38%. It is observed that if no new policy is created in the organization's program, agility will be reduced during five years, so to maintain SC agility, the mentioned policies have been implemented and simulated. The results in Fig. 13 show what happens if any of these scenarios occur if these policies are implemented. Table 2 shows the results of the implementation of these policies. According to the results of the implementation of these policies at the time of occurrence of each scenario, it has been shown that implementing policies at the same time increase cooperation and intra-group partnerships, investment, HCD and accelerates the completion of ongoing projects based on organizational priorities. For purpose of the study, the agility of the SC will be maintained for 5 years, due to the stated benefits (increased inflation and the discovery of COVID-19), and even increased slightly.

• How to better determine the amount of capital and workforce required for ongoing projects.

Response: We really appreciate this valuable and constructive comment. The system dynamics model determined the amount of output based on internal equations. Also, the equations work based on historical data we collected from case studies. In this study, the better amount of each critical variable of each scenario -policies are in Table 2.

---

## [Editor Report · Decision Letter 1]

16 Aug 2023

System Dynamics Model for an Agile Pharmaceutical Supply Chain During COVID‑19 Pandemic in Iran

PONE-D-23-03622R1

Dear Dr. Hamzehlou,

We’re pleased to inform you that your manuscript has been judged scientifically suitable for publication and will be formally accepted for publication once it meets all outstanding technical requirements.

Kind regards,

Saad Ahmed Javed, Ph.D

Academic Editor

PLOS ONE

Additional Editor Comments (optional):

The manuscript is acceptable.
---

## [Editor Report · Acceptance letter]

24 Aug 2023

PONE-D-23-03622R1 

System Dynamics Model for an Agile Pharmaceutical Supply Chain During COVID‑19 Pandemic in Iran 

Dear Dr. Hamzehlou:

I'm pleased to inform you that your manuscript has been deemed suitable for publication in PLOS ONE. Congratulations! Your manuscript is now with our production department. 

Kind regards, 

on behalf of

Dr. Saad Ahmed Javed 

Academic Editor

PLOS ONE